# Simulating quantum light propagation through atomic ensembles using matrix product states

Marco T. Manzoni[1], Darrick E. Chang[1] & James S. Douglas [1]

A powerful method to interface quantum light with matter is to propagate the light through an ensemble of atoms. Recently, a number of such interfaces have emerged, most prominently Rydberg ensembles, that enable strong nonlinear interactions between propagating photons. A largely open problem is whether these systems produce exotic many-body states of light and developing new tools to study propagation in the large photon number limit is highly desirable. Here we provide a method based on a "spin model" that maps quasi one-dimensional (1D) light propagation to the dynamics of an open 1D interacting spin system, where all photon correlations are obtained from those of the spins. The spin dynamics in turn are numerically solved using the toolbox of matrix product states. We apply this formalism to investigate vacuum induced transparency, wherein the different photon number components of a pulse propagate with number-dependent group velocity and separate at output.

---

[1] ICFO-Institut de Ciencies Fotoniques, The Barcelona Institute of Science and Technology, Castelldefels, 08860 Barcelona, Spain. Correspondence and requests for materials should be addressed to J.S.D. (email: james.douglas@icfo.eu)

Atomic ensembles are a very successful platform used to couple input light to atomic degrees of freedom, allowing the development of new quantum technologies. Historically, the weak optical nonlinearities associated with atomic ensembles have allowed most of the processes of interest, such as quantum memories for light[1], to be describable within a limited realm of classical linear optics or Gaussian quantum states[2]. More recently, it has become possible to engineer strong interactions between photons in atomic ensembles and thereby realize highly non-Gaussian states. Under weak field inputs, for example, phenomena such as photon blockade[3] or two-photon bound states[4] in atomic Rydberg gases[5-7] have been experimentally demonstrated. A number of other systems, such as photonic waveguides coupled to atoms[8-14] or "artificial" atoms such as superconducting qubits and quantum dots[15-18] also show potential to realize similar physics.

A major question of interest is what occurs in such systems at higher field inputs. In particular, it is expected that strong interactions might lead to interesting many-body phenomena involving photons, such as photon crystallization (illustrated schematically in Fig. 1a). To address this question theoretically seems challenging: the systems are out of equilibrium, being driven by an external laser source; are open, where spontaneous decay of atoms leads to losses; and have long range interactions between atoms mediated by the exchange of photons. Some progress has been made in limiting regimes, where for example effective theories can emerge under certain approximations[11, 12, 19-26]. While these effective theories provide useful insights, it would also be highly desirable to develop numerical methods with minimal approximations to verify these models and investigate regimes where the approximations break down, potentially revealing new physics as a result.

Currently numerical techniques are quite limited. For example, the standard approach is to describe the atom-light dynamics using Maxwell-Bloch equations[27, 28], which are solved by discretizing the atomic and photonic fields[3], i.e., by modeling the continuum in which fields propagate by a finite number $M$ of boxes, as depicted in Fig. 1b. To describe a state with up to $n$ photons in each box then requires a Hilbert space with dimension of at least $(n+1)^M$ (plus any atomic degrees of freedom associated with each box). In practice, this has limited numerical simulations to two[3, 4, 20, 29, 30] or three[23] total excitations over the entire system, and excluding the full effects of dissipation (in particular, either quantum jumps in wave function evolution, or population recycling terms in density matrix evolution).

Here, we show instead that simulating the high-photon number limit is possible by mapping the propagation problem onto the dynamics of a one-dimensional (1D) open "spin" model, which can be solved using the powerful matrix product state (MPS) ansatz[31, 32]. The essence of the spin model is that in light propagation through an ensemble, the only independent degrees of freedom are the atomic internal states ("spins"), where the light fields mediate interactions between the atoms. A common theoretical approach is then to integrate out the fields, reducing the description to a problem of $N$ interacting spins ($N$ being the number of atoms)[33-39]. Furthermore, many light propagation experiments are quasi-one dimensional, where the input and output light are in a single transverse mode[2]. In this case, a simpler model capturing the same physics is a 1D spin chain coupled by the modes of a 1D photonic waveguide, as shown in Fig. 1c.

Our model further takes advantage of the fact that typically the number of spins in the chain does not appear independently, but rather macroscopic observables such as optical depth depend on the product of atom number and the atom-photon interaction probability. By tuning this probability to be large, a relatively

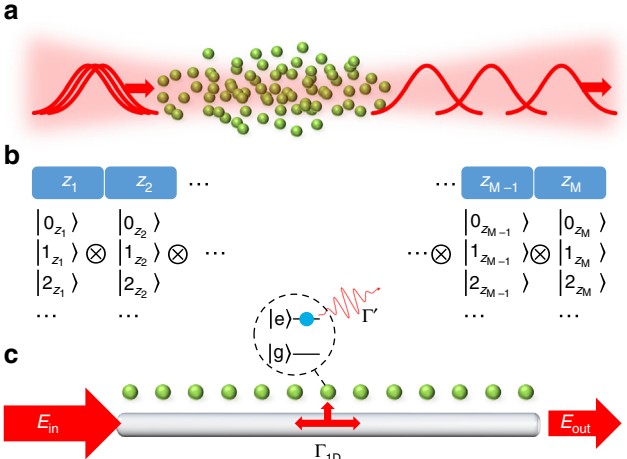

**Fig. 1** Photon propagation in atomic ensembles. **a** Propagation through atomic ensembles with strong inter-atomic interactions can lead the quantum properties of the light to be strongly modified, represented here as crystalline-like output of photons from the medium. **b** In a typical approach to treat such quantum light propagation numerically, space is broken up into $M$ discrete units centered at positions $z_j$, with corresponding local Fock basis states $|0_{z_j}\rangle, |1_{z_j}\rangle, \ldots$ for the photons. The equations of motion for the field may be solved on this space, where the growth of the Hilbert space with the number of Fock components on each site typically restricts calculations to a maximum of two or three photons in the entire system. **c** Here instead we model the quasi-1D propagation problem by a 1D waveguide coupled to $N$ atoms. In this case the degrees of freedom associated with the light field are integrated out, to produce an effective model consisting of an interacting "spin" chain (with the spins being associated with atomic internal states). Output fields are then directly calculated from the spin dynamics using an input-output relation and the Hilbert space is reduced to the atomic one, which can be efficiently treated using matrix product states. As discussed in the Results section, the atoms are coupled to the waveguide modes with strength $\Gamma_{1D}$ and for the simple case of two-level atoms the excited state $|e\rangle$ may also decay into modes outside of the waveguide at rate $\Gamma'$

small ($\sim 10^2$) number of atoms is sufficient to model most atomic ensemble experiments. We can then use the MPS technique from condensed matter physics to numerically simulate the system, which is well adapted to treating chains of hundreds of spins. This technique depends on the fact that many of the quantum states we encounter in reality do not have large amounts of entanglement and are confined to a small portion of the in-principle exponential Hilbert space, allowing for a more efficient state representation. While the amount of entanglement present in atom-light interfaces is generally an unstudied problem, we give heuristic arguments below why we expect the MPS ansatz to work for such systems. As a benchmark of our model, we use it to simulate pulse propagation in the case of vacuum induced transparency[40, 41], which is one of the few cases where many-photon propagation is qualitatively understood[30, 42].

## Results

**1D spin model of light propagation.** While there are some phenomena in atomic ensembles that are truly three-dimensional, such as radiation trapping[43] and collective emission at high densities[44-46], within the context of generating many-body states of light, the problems of interest largely involve quasi one-dimensional propagation[19-24, 26]. Indeed a typical experimental design is to input light in a single transverse mode and detect the light in the same mode after it traverses the ensemble. The standard approach to describe light propagation in such a system

is to use Maxwell–Bloch equations[27, 28] in their one-dimensional, paraxial form[2–4, 20, 47–49]. There, the electric field operator can be decomposed as the sum of a forward propagating mode that travels in the direction of the input, taken here to be the positive $z$ direction, and a backward propagating one, $E(z, t) = E_+(z, t) + E_-(z, t)$. These have propagation equations,

$$\left(\frac{1}{c}\frac{\partial}{\partial t} \pm \frac{\partial}{\partial z}\right) E_\pm(z, t) = i\sqrt{\frac{\Gamma_{1D}}{2}} P_{ge}(z, t), \quad (1)$$

determined by the atomic polarization density operator $P_{ge}(z)$. For concreteness, we assume that the probe field couples to a single dipole-allowed transition from atomic ground state $|g\rangle$ to excited state $|e\rangle$, however, it is straightforward to modify these equations to account for additional atomic levels and transitions, driving fields, and interactions.

The atomic polarization density, on the other hand, is driven by the field, and obeys an optical Bloch equation

$$\partial_t P_{ge}(z, t) = -i\left(\omega_{eg} - i\Gamma'/2\right) P_{ge}(z, t) \\ + i\sqrt{\frac{\Gamma_{1D}}{2}}\left[P_{gg}(z, t) - P_{ee}(z, t)\right] E(z, t) + F(t), \quad (2)$$

where $\omega_{eg}$ is the atomic transition frequency, $P_{ee,gg}$ are the excited and ground state populations, and $F$ describes the quantum noise associated with decay rate $\Gamma'$. Here we have introduced the coupling rate $\Gamma_{1D}$ of an individual atom to the one-dimensional input mode. In principle, this rate can vary with $z$ depending on the details of this mode, but for notational simplicity we assume here that it is constant. In this standard formulation of the Maxwell-Bloch equations, it should be noted that the interaction of the atoms with the remaining continuum of three-dimensional modes is reduced to an independent emission rate $\Gamma'$, meant to approximately capture scattering of photons out of the transverse mode of interest. The question of when this approximation breaks down is quite complicated and rich[20, 50, 51] and will not be discussed here; in any case, Eqs. (1, 2) are widely accepted as the standard model for quasi-1D light propagation through atomic ensembles.

Eqs. (1, 2) represent an open, interacting quantum field theory, for which a general solution is unknown. The complexity is reduced in ensembles that lack strong non-linearities, where for example one can linearize the atomic system, such that the resulting joint quantum state of matter and light is Gaussian[2]. However, in the highly non-linear ensembles that are interesting for many-body physics we are typically restricted to solving numerically the Maxwell-Bloch equations by discretizing the fields, which, as mentioned in the introduction, is not feasible for more than a few photonic excitations.

Here instead, we take advantage of the fact that the Maxwell-Bloch equations presented above can also formally arise from a simple toy model of atoms coupled to a 1D waveguide[36, 52–55]. In particular, one can consider a model of atoms coupled to a bi-directional waveguide of infinite bandwidth, and frequency-independent propagation speed $c$ and coupling strength. In that case, the propagation equations of the forward and backward going modes are exactly those in Eq. (1), where $P_{ge}(z) = \sum_{j=1}^{N} \sigma_{ge}^j \delta(z - z_j)$ for $N$ atoms with positions $z_j$. These equations can then be formally integrated giving a solution for the field as the sum of the input field $\mathcal{E}_{in}(z, t)$ and the field radiated by the atoms[36–39],

$$E(z, t) = \mathcal{E}_{in}(z, t) + i\sqrt{\frac{\Gamma_{1D}}{2}} \sum_{j=1}^{N} e^{ik_0|z-z_j|} \sigma_{ge}^j(t). \quad (3)$$

In the above equation the propagation of the field from the atomic position $z_j$ to $z$ leads to a phase factor determined by the wavevector $k_0 = \omega_{eg}/c$. On the other hand, the time delay in free-space propagation is neglected and the field sees the response of an atom at another point instantly. That is, any pulse delay that arises is due to the atomic dispersion itself. This is justified when the free-space propagation time over the length of the system $L$ is much smaller than the time scale characterising the atomic evolution, e.g., when $L/c \ll 1/\Gamma'$, a condition easily satisfied in atomic ensemble experiments[36]. In limiting cases, this approximation can be further validated by solving for dynamics exactly and seeing that the results are the same[56].

Removing time retardation provides a drastic simplification of the problem as the equations of motion of the atoms and fields are now all local in time. Indeed, inserting Eq. (3) into the Heisenberg equation for the atomic coherences, one finds that the dynamics of the atoms depend only on the input field and on the state of the other atoms at the same time. Part of the system dynamics can then be derived from an atomic interaction Hamiltonian[36, 54],

$$H_{dd} = \frac{\Gamma_{1D}}{2} \sum_{j,l=1}^{N} \sin\left(k_0|z_j - z_l|\right) \sigma_{eg}^j \sigma_{ge}^l \quad (4)$$

which describes the process of emission by an excited atom at $z_j$ into the waveguide, and the subsequent absorption of that photon by a ground-state atom at $z_l$. These effective atom-atom interactions also lead to collective spontaneous emission, described in the master equation by the Lindblad operator

$$\mathcal{L}_{dd}[\rho] = \frac{\Gamma_{1D}}{2} \sum_{j,l=1}^{N} \cos\left(k_0|z_j - z_l|\right) \\ \times \left(2\sigma_{ge}^j \rho \sigma_{eg}^l - \sigma_{eg}^j \sigma_{ge}^l \rho - \rho \sigma_{eg}^j \sigma_{ge}^l\right). \quad (5)$$

In addition, we can add a phenomenological independent decay rate $\Gamma'$ as in the Maxwell-Bloch equations, which corresponds to scattering out of the quasi-1D input mode. This is described by the locally acting Lindblad operator $\mathcal{L}_{spont}[\rho] = \Gamma'/2 \sum_{j=1}^{N} (2\sigma_{ge}^j \rho \sigma_{eg}^j - \sigma_{eg}^j \sigma_{ge}^j \rho - \rho \sigma_{eg}^j \sigma_{ge}^j)$.

The coupling of the atoms to the input field is given by $H_{drive} = -\sqrt{\Gamma_{1D}/2} \sum_{j=1}^{N} (\mathcal{E}_{in}(t, z_j)\sigma_{eg}^j + \text{H.c.})$. In the following we will consider the case most common in experiments of a coherent state input, where $\mathcal{E}_{in}$ can be treated as a classical field[57] and we neglect the associated quantum noise term, as this does not contribute to the normally ordered correlation functions of the output field that we will be interested in. The output field itself is the field measured past the position of the last atom, $E_{out}(t) = E(z_N^+, t)$ given by Eq. (3), which is completely determined by the solution of the driven spin system and the input.

In the model above, the coupling of the atoms to the waveguide and the positions of the atoms must be chosen carefully to reproduce phenomena associated with free-space ensembles. In particular, as we discuss below, our numerical calculations are facilitated by choosing ratios of $\Gamma_{1D}/\Gamma' \sim 1$. It is known that for a weak resonant input field, a single two-level atom can produce an appreciable reflectance of $\Gamma_{1D}^2/(\Gamma_{1D} + \Gamma')^2$[52, 53]. The reflectance can be further enhanced if multiple atoms are placed on a lattice with lattice constant defined by $k_0 a = \pi$, in which case the reflectance from individual atoms constructively interferes[54, 58, 59]. While it is possible to observe similar effects in atomic ensembles[60, 61], this situation is atypical and will not be discussed further here. To reproduce the typical case in atomic ensembles where reflection is negligible, we choose a waveguide spacing of

$k_0 a = \pi/2$, in which case reflection from different atoms in the lattice destructively interferes.

In this configuration, the 1D waveguide model reproduces one of the key features of an atomic ensemble, that of decay of the transmitted field with increasing optical depth. If we consider the transmittance $T = \langle E_{out}^\dagger E_{out} \rangle / |\mathcal{E}_{in}|^2$, then for a resonant weak coherent state input we find in the 1D waveguide model $T = \exp(-OD)$, where the optical depth is $OD = 2N\Gamma_{1D}/\Gamma'$ for $\Gamma_{1D} \lesssim \Gamma'$ [12]. Since $OD \lesssim 10^2$ in realistic atomic ensembles of $\sim 10^6$ weakly coupled atoms, by artificially choosing $\Gamma_{1D} \sim \Gamma'$, the same optical depth is achieved with just tens or hundreds of atoms. At the same time, the essential properties of most quantum nonlinear optical phenomena are believed to depend only on optical depth[47, 62, 63], or on a limited number of other parameters where atom number does not appear independently (such as the optical depth per blockaded region in a Rydberg gas[6, 26, 29]). By matching these parameters using a much smaller number of atoms, we can then model the physics of interest in 3D atomic ensembles. The possibility that artificial effects (such as saturation) arise from low atom number can be eliminated by numerically checking that observables converge with increasing $N$ while, e.g., decreasing $\Gamma_{1D}$ in proportion to keep the key parameters fixed.

While our model can be used to reproduce the macroscopic observables of light propagation in a traditional atomic ensemble, we also note that it quantitatively captures the microscopic details of experiments where atoms or other quantum emitters couple to 1D channels. This includes atoms coupled to nano-fibers ($\Gamma_{1D}/\Gamma' \sim 0.05$)[13] or photonic crystals ($\Gamma_{1D}/\Gamma' \sim 1$)[14], or "artificial" atoms such as superconducting qubits or quantum dots coupled to waveguides ($\Gamma_{1D}/\Gamma' \gg 1$)[15–18]. In these cases, our model is valid when the spacing between the atoms is of the order of the wavelength of the light or larger, for smaller atomic distances additional effects can occur[51, 64].

**Simulations using matrix product states**. Using the 1D spin model described above significantly reduces the size of the Hilbert space required to simulate the light propagation problem, but the dimension still grows exponentially with atom number. This growth can be avoided in the case where the input field is sufficiently weak that the Hilbert space can be truncated to a maximum number of total excitations likely to be found in the system[12, 36]. In the more general case, where many-photon effects are important, the full Hilbert space may be treated numerically for around 10–20 atoms depending on the size of the single-atom Hilbert space dimension $d$. Going beyond this requires some reduction of the Hilbert space and here we choose to use matrix product states, which have been successfully used in condensed matter to model a wide variety of 1D interacting spin systems[31, 32].

The key idea behind MPS is to write the quantum state of the spin chain in a local representation where only a tractable number of basis states from the full Hilbert space is retained. In the case of time evolution, these basis states are updated dynamically in order to have optimum overlap with the true state wave function. In particular, the wave function of a many-body system $|\psi\rangle = \sum_{\sigma_1,\dots,\sigma_N} \psi_{\sigma_1,\sigma_2,\dots,\sigma_N} |\sigma_1, \sigma_2, \dots, \sigma_N\rangle$ can be represented by reshaping the $N$-dimensional tensor $\psi_{\sigma_1,\sigma_2,\dots,\sigma_N}$ into a matrix product state of the form

$$|\psi\rangle_{MPS} = \sum_{\sigma_1,\dots,\sigma_N} A^{\sigma_1} A^{\sigma_2} \dots A^{\sigma_N} |\sigma_1, \sigma_2, \dots, \sigma_N\rangle, \qquad (6)$$

where $\sigma_j$ represent the local $d$-dimensional Hilbert space of the atoms, e.g., $\sigma_j \in \{|e\rangle, |g\rangle\}$ for two-level atoms. Each site $j$ in the spin chain has a corresponding set of $d$ matrices, $A^{\sigma_j}$, and by taking the product of these matrices for some combination of $\sigma_j$'s

we then recover the coefficient $\psi_{\sigma_1,\sigma_2,\dots,\sigma_N}$. The matrices have dimensions $D_{j-1} \times D_j$ for the $j$th site ($D_0 = D_{N+1} = 1$), which are referred to as the bond dimensions of each site. We also define $D = \max_j \{D_j\}$ as the maximum bond dimension of the state $|\psi\rangle_{MPS}$. This representation is completely general, and as such the bond dimensions grow exponentially in size for arbitrary quantum states. In certain circumstances, however, the bond dimension $D$ needed to approximate a state well might grow more slowly with $N$ due to limited entanglement entropy, which enables MPS to serve as an efficient representation.

For example, this forms the underlying reason for the efficiency of density-matrix renormalization group algorithms for computing ground states of 1D systems with short-range interactions[65]. A priori, for our system involving the dynamics of an open system with long-range interactions, we know of no previous work that makes definitive statements about the scaling of $D$. However, we can provide some intuitive arguments that MPS should work well (at least without additional interactions added to the system). First, we note that although the dipole-dipole interaction term in Eq. (4) appears peculiar, being infinite-range and non-uniform, it conserves excitation number. For a single excitation, it simply encodes a (well-behaved) linear optical dispersion relation that propagates a pulse from one end of the atomic system to the other[12], and thus does not add entanglement to the system. While the spin nature in principle makes the atoms nonlinear, thus far in atomic ensemble experiments the strength of nonlinearity arising purely from atomic saturation remains very small at the level of single photons, and thus one can hypothesize that only a small portion of the Hilbert space is explored.

Once extra interactions are added, at the moment the scaling of $D$ must be investigated on a case-by-case basis. However, generically one expects that the system has a memory time corresponding roughly to the propagation time of a pulse through the length of the system. Thus, if the system is driven continuously, it should generally reach a steady state over this time and there will not be an indefinite growth of entanglement. In the case of pulsed input, the number of photons in the pulse limits entanglement and it is possible to establish upper bounds on the bond dimension required as we discuss in the Supplementary Note 1. For arbitrary $n$-photon wave functions the bound may still scale exponentially in the number of photons $N^{m/2}$, however in the case of vacuum induced transparency that we investigate as a benchmark the scaling is instead approximately quadratic in the number of photons.

In our MPS treatment of the spin model we adopt a quantum jump approach to model the time dynamics of the master equation[66], which has been successfully applied to many-body dissipative systems[67, 68]. As we describe in more detail in the Methods, this method decomposes the master equation evolution into an ensemble of quantum trajectories, which are formed by deterministically evolving pure states under an effective Hamiltonian $H_{eff}$ and stochastically applying quantum jumps to the system. As an aside, however, we note that MPS-based techniques for evolution of density matrices have also been developed[69–72]. Whether and when such techniques out-perform quantum jump methods for our problem is likely a subtle question, which will be explored in more detail in future work.

To numerically simulate the spin model there are then four essential manipulations of the MPS as illustrated in Fig. 2 and described in greater detail in the Methods. The first is (a) deterministic evolution of the MPS over a small discrete time step $\delta t$. Here an approximation of the time evolution operator $e^{-iH_{eff}\delta t} \approx 1 - iH_{eff}\delta t$ is applied to the MPS representation of the state. This is achieved by expressing the operator as a matrix product operator (MPO), a generalization of MPS to operators.

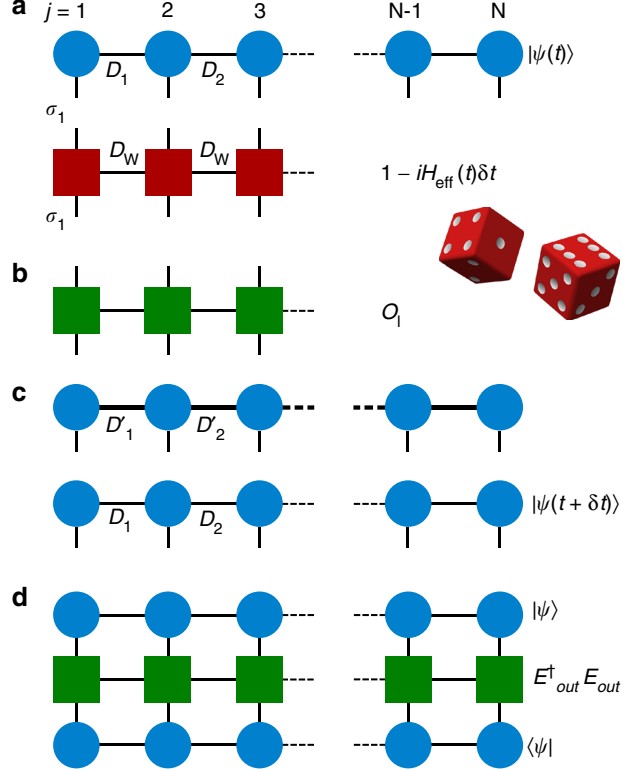

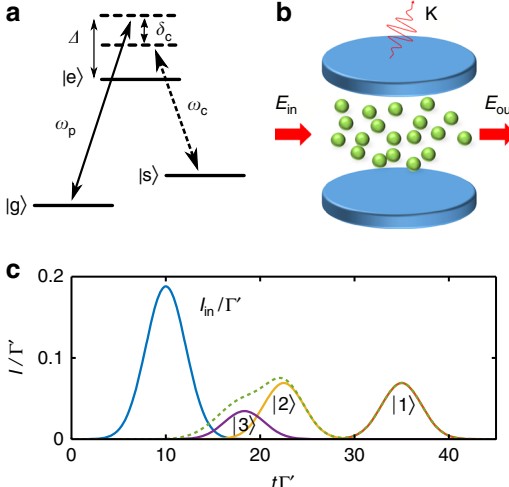

**Fig. 2** Schematic of MPS operations. **a** Deterministic time evolution of MPS. The initial state $|\psi(t)\rangle$ in MPS form is presented pictorially as a tensor network, where the circles represent the set of local matrices $A^{\sigma_j}$ on each site $j$. The lines or bonds joining the circles represent the contraction of these local tensors to give the state $|\psi(t)\rangle$, where the bonds have dimension $D_j$. The open ended lines correspond to the local $d$-dimensional Hilbert space of the atoms $\sigma_j$. The deterministic evolution is then found by contracting these open connectors with those of the MPO representing $e^{-iH_{\text{eff}}\delta t} \approx 1 - iH_{\text{eff}}\delta t$, shown as a tensor network of red squares. **b** Quantum jumps. After each deterministic evolution a random number generator is used to decide whether quantum jumps should be applied to the wave function. This is achieved by applying the MPO corresponding to a quantum jump $O_l$, shown as a tensor network of green squares. **c** After the application of the time evolution or jump MPOs the resulting MPS has larger bond dimension, e.g., $D_1' = D_W \times D_1$, and is compressed, typically back to the original bond dimension, although this can be increased if the compression produces a large error. **d** Measurement of observables. At any time we may measure an observable by sandwiching the corresponding MPO, here for example $E^\dagger_{\text{out}}E_{\text{out}}$, between the MPSs representing $|\psi\rangle$ and $\langle\psi|$, so that the corresponding tensor contraction yields $\langle\psi|E^\dagger_{\text{out}}E_{\text{out}}|\psi\rangle$

After this deterministic evolution the MPS then undergoes (b) stochastic quantum jumps that account for dissipation, realised again by applying MPOs to the MPS, where this time the MPOs correspond to the quantum jump operators $O_l$. In each case, after applying an MPO (with corresponding bond dimension $D_W$ defined in the same way as for an MPS) to an MPS, the MPS bond dimension increases. The state must then be (c) compressed to constrain the growth of its representation in time. At any time step we may then (d) calculate observables, such as the output field, given the MPS representation of a state and the MPO corresponding to the observable. The steps are then repeated to obtain the full time evolution.

**Vacuum induced transparency**. The model introduced above gives a powerful and flexible algorithm for simulating the interaction of light with atomic ensembles in the multi-photon limit.

**Fig. 3** Vacuum induced transparency. **a** Three-level atoms, where the transition $|s\rangle$-$|e\rangle$ is coupled to a control or cavity field with frequency $\omega_c$, allow for transparent propagation of probe photons ($\omega_p$) in EIT or VIT. **b** In VIT, an atomic ensemble is trapped inside an optical cavity where the atoms couple both to the probe field $E_{\text{in}}$ and to a cavity mode, which is initially in its vacuum state. Photons in the cavity have an associated decay rate $\kappa$ from transmission through the mirrors. **c** Idealized time-dependent transmission of a coherent pulse with average number of photons equal to one through a VIT medium. Here, the intensity (photons per unit time) is normalized by the single-atom decay rate $\Gamma'$ into free space, while the time $t$ is also normalized by $\Gamma'$. In the case where all loss mechanisms are ignored, as well as the effect of pulse distortion on entry and exit from the atomic ensemble, the individual Fock number state components $|n\rangle$ of the input pulse (blue) propagate through the medium with group velocity $v_n \propto n$. This leads to separation of the one- (red), two- (yellow) and three-photon (violet) components of the output field, and a total output intensity shown by the green dashed line. We have taken $v_1 = 4a\Gamma'$ and the medium has a length $L = 100a$

To demonstrate the utility of this approach we now investigate the phenomenon of vacuum induced transparency (VIT)[40]. This example also serves to benchmark our method, as exact solutions for non-trivial multiphoton behavior are not available, while in the case of VIT at least the qualitative nature of the system dynamics is understood.

VIT is closely related to the effect of electromagnetically induced transparency (EIT)[1], which occurs in three-level atomic media. In a two-level medium incoming probe light that couples resonantly to an atomic transition $|g\rangle$-$|e\rangle$ is absorbed and scattered by the atoms into other directions, leading for example to the strong attenuation in the linear transmittance for high optical depth, $T = \exp(-\text{OD})$. In EIT, an additional metastable level $|s\rangle$ (Fig. 3a) is also coupled to the excited state by a classical control field with Rabi frequency $\Omega$, allowing probe photons to couple with spin-wave excitations from state $|g\rangle$ to $|s\rangle$, forming so-called "dark-state polaritons". The coupling to the spin wave leads to a strongly reduced group velocity relative to free space ($v_g = \Omega^2 a/(2\Gamma_{1D})$ for a waveguide with spin chain lattice constant $a$[12]), while the absence of population in $|e\rangle$ enables a pulse to propagate with minimal attenuation.

In VIT the control field is replaced by strong coupling of the atoms to a resonant cavity mode as shown in Fig. 3b[40, 73], which is described by the Hamiltonian $H_{\text{cav}} = g \sum_{j=1}^{N} (\sigma^j_{es}b + \text{H.c.})/2$ in the case of uniform coupling $g$ to a cavity mode with annihilation operator $b$. Here even when the cavity is empty the atomic medium can become transparent as vacuum Rabi oscillations transfer population from state $|e\rangle$ to $|s\rangle$[41]. The

propagation of light in the system then takes on the nature of the non-linear coupling of the atoms to the cavity. Specifically, the formation of a spin wave from $n$ probe photons is accompanied by the excitation of the same number of cavity photons, which produce an effective control field strength of $\sqrt{n}g$. Since in EIT the group velocity of the light is determined by the control field, where $v_g \propto \Omega^2$, the group velocity in VIT becomes number dependent $v_n \sim ng^2a/(2\Gamma_{1D})$[30, 42]. Fock states $|n\rangle$ input into the system are then expected to propagate at $v_n$.

On the other hand, a coherent state $|\alpha\rangle$ that has average number of photons $|\alpha|^2$ is a superposition of Fock states, where $n$ photons are present with probability $e^{-|\alpha|^2}|\alpha|^{2n}/n!$. Input into the VIT medium, these components are then expected to spatially separate due to their different propagation velocities, given sufficient optical depth. The output intensity can then be calculated naively by simply delaying the input Fock components by a time $\tau_n = L/v_n$, where $L$ is the length of the atomic medium. The output intensity in time resulting from such a toy model is shown in Fig. 3c, for a coherent state input pulse with average number $\langle n_{\text{pulse}} \rangle = 1$. We have taken the system length to be $L = 100a$ and the single photon velocity $v_1 = 4a\Gamma'$, which results for example from taking $g = 4\Gamma'$ and $\Gamma_{1D} = 2\Gamma'$ in which case the system's optical depth is 400. We note that the experimental conditions needed to observe photon number separation in VIT are difficult to achieve[41], and thus our parameters are chosen to observe the desired effect, rather than correspond to a given experiment.

A plot similar to Fig. 3c was given in a previous theoretical treatment of VIT[42], as at that time it was unknown how to calculate observables in the presence of losses and spatio-temporal effects, such as occurring from pulse entry and exit from the atomic medium. More recently, VIT has also been studied numerically in the weak-field limit using the space discretization technique schematically illustrated in Fig. 1b[30]. In the weak field limit, only the single photon manifold contributes to the output intensity and the higher number components are only visible in higher order correlation functions like $g^{(2)}$. This also means that quantum jumps have a negligible effect on the system dynamics, and they were neglected in the calculations. In more general circumstances, using MPS simulations, we will show that the effects of quantum jumps and pulse distortion can have a significant effect on the output field.

For concreteness, we take input pulses with central frequency $\omega_{P_{1/4}}$ and Gaussian envelope $\mathcal{E}_{\text{in}}(t) = \alpha \left( \pi\sigma_t^2/2 \right)^{1/4} \exp \left( -(t - t_0)^2/\sigma_t^2 \right)$, which have an average photon number of $\langle n_{\text{pulse}} \rangle = |\alpha|^2 \sim 1$. The average photon number chosen is not due to any intrinsic limitation coming from the MPS method itself, but rather because in VIT the spatial separation is largest for the Fock components with low photon number (Fig. 3c) and with $|\alpha|^2 = 1$ the single photon and two photon components of the coherent state give an equal contribution to intensity emphasizing this effect. In this case, number states with three or more photons make up 8% of the input state and constitute 26% of the input intensity due to their high photon number.

To treat VIT, we include in the spin model formalism the atomic part $H_0$ of the total effective Hamiltonian (Eq. (8) in Methods section),

$$H_0 = -\sum_{j=1}^{N} \left( \Delta + i\frac{\Gamma'}{2} \right) \sigma_{ee}^j - \left( \delta_c + i\frac{\kappa}{2} \right) b^\dagger b + \frac{g}{2} \sum_{j=1}^{N} \left( \sigma_{es}^j b + \text{H.c} \right).$$

(7)

Here $\Delta = \omega_p - \omega_{eg}$ is the detuning of the probe light from the

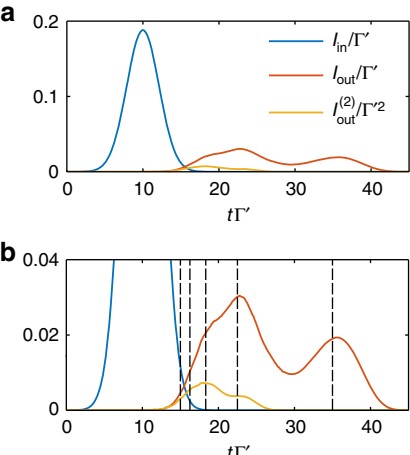

**Fig. 4** VIT output field. **a** Pulse propagation in a VIT medium with optical depth OD = 400, simulated using $N = 100$ atoms coupled to a 1D-waveguide, and averaged over 20000 quantum trajectories. Input of a coherent pulse with $|\alpha|^2 = 1$ (blue) results in an output intensity $I_{\text{out}}(t)$ (red) with two main peaks. Also plotted is the second-order correlation function $I_{\text{out}}^{(2)}(t, t)$ (yellow). **b** Zoom of the plot above, with dashed lines showing the expected positions of pulses delayed by $\tau_n$, for $n = 1, ..., 5$. Simulation parameters are $\Gamma_{1D} = 2\Gamma'$, $\Delta = \delta_c = 0$, $g = 4\Gamma'$, $\kappa = 0.03\Gamma'$, $\sigma_t = 3/\Gamma'$ and $t_0 = 10/\Gamma'$. We chose $D = 50$ and $\delta t = 0.01/\Gamma'$ where convergence was observed for all observables of interest (see Methods)

$|e\rangle$-$|g\rangle$ transition frequency, $\delta_c = \omega_p - \omega_c - \omega_{sg}$ is the VIT two-photon detuning and $\kappa$ is the decay rate of the cavity mode. In what follows we assume both the probe and cavity are resonant with their respective transitions, so that $\Delta = \delta_c = 0$. Dissipation via the various loss channels is then included through quantum jump operators, where the collective emission into the waveguide is described by $O_\pm$ as detailed in the Methods. The jump operator corresponding to cavity decay is $O_c = \sqrt{\kappa}b$ and we assume that the atomic excited state can decay via free-space spontaneous emission into either state $|g\rangle$ or $|s\rangle$ (taking these decay rates to be equal for simplicity), leading to $2N$ jump operators $O_{j,ge} = \sqrt{\Gamma'/2}\sigma_{ge}^j$ and $O_{j,se} = \sqrt{\Gamma'/2}\sigma_{se}^j$. The cavity mode is represented in our MPS treatment by an additional site in our spin chain, which can support up to $n_c$ bosonic excitations. In the simulations we present here we have taken $n_c = 10$ and observe no difference in observables if $n_c$ is increased.

In Fig. 4a, b, we show the time-dependent output pulse intensity $I_{\text{out}}(t) = \langle E_{\text{out}}^\dagger(t) E_{\text{out}}(t) \rangle$ calculated from an MPS simulation of 100 atoms and an input pulse with $|\alpha|^2 = 1$. We also show the zero-delay second-order correlation function $I_{\text{out}}^{(2)}(t, t) = \langle E_{\text{out}}^\dagger(t) E_{\text{out}}^\dagger(t) E_{\text{out}}(t) E_{\text{out}}(t) \rangle$. In the output intensity two main peaks are observed, where the first peak in time ($t\Gamma' \sim 23$) is due to photon number components with two or more photons, while the last peak ($t\Gamma' \sim 36$) is associated with the slow propagation and exit of the single-photon component. That the most delayed part contains only single photons can be confirmed by looking at the second order correlation function which is only non-zero in the first part of the pulse. In Fig. 4b, we see good agreement between the features of the numerical pulse shape and the expected group velocity for each part of the pulse (compare with Fig. 3c), where the vertical black dashed lines represent the expected times for the peaks of the Fock state components, that is, with delays $\tau_n$.

Compared with the ideal picture in Fig. 3c, where a clean separation is seen between one and two photons, one can see that the full simulation produces a much larger intensity between the one- and two-photon peaks. We now show how the trajectories

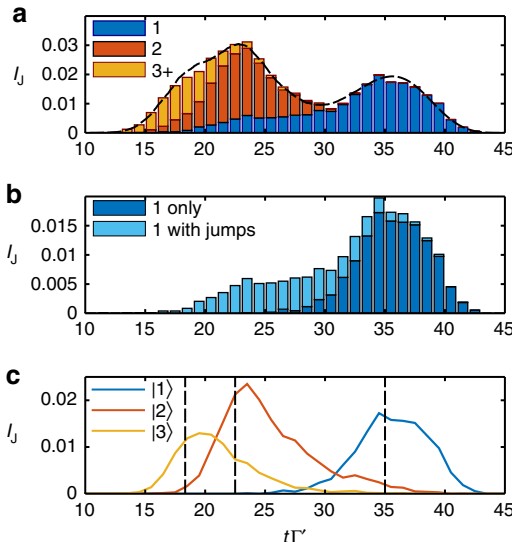

**Fig. 5** VIT quantum jumps statistics. **a** Stacked bar graph of quantum jumps into the output channel over the 20000 quantum trajectories used in Fig. 4. The height of each bar, $I_J$, is the average number of jumps occurring in the time bin defined by the bar's width, in this case $1/\Gamma'$. The bars are then divided into three categories by classifying each jump according to how many jumps into the output channel occur for a particular trajectory (1, 2, or 3 or more). Jumps from trajectories where there are a higher number of photons emitted into the output channel are seen to occur earlier. For comparison the dashed black line shows the output intensity from Fig. 4. **b** Stacked bar graph for quantum jumps from trajectories where only a single photon is detected in the output of the waveguide. These jumps are then divided into jumps that are not accompanied by any other jump into other channels, and those that are. We see that the tail of photons detected earlier are due to trajectories where 2 or more photons entered the medium but all but one were lost into other channels. **c** Post-selection of trajectories to find evolution for Fock state input. By selecting only trajectories where there were a total of 1 (blue), 2 (red), or 3 (yellow) jumps into any channel we can reconstruct the intensity output for the corresponding input Fock state

from the MPS simulations can be further filtered and analyzed, to gain insight about the underlying physics. In particular, we find that quantum jumps play a key role in blurring the separation between the different number components in the output, even for the very good system parameters that we have chosen (OD = 400, $g/\kappa \sim 130$). An intuitive picture of how the blurring occurs can be gained by considering two photons that enter the medium, and initially propagate at a velocity $v_2 = 2v_1$. During evolution, this state may decay via spontaneous emission into free space and leave behind a single photon propagating in the medium, at which point the group velocity is slowed to $v_1$. This change in group velocity can happen at any point in the system and leads to single photons that arrive at the output earlier than expected if just a single-photon Fock state was input into the system, destroying the perfect separation of the single photon output from the two photon component.

We can quantify this behavior by analyzing the quantum jumps that happen in our simulations, where due to the choice of physical jump operators discussed in the Methods, the total number of jumps in a given trajectory corresponds to the number of photons emitted from the system. Furthermore, the type of jumps (and thus the emission channel) can be explicitly tracked, between free-space loss, cavity loss, or detection in the waveguide output. In Fig. 5a, we show a histogram of the average number of jumps into the output waveguide channel versus time for the

20000 trajectories used to produce Fig. 4. This provides an alternative way (compare to Fig. 4) to calculate the intensity, as would be done in an experiment where detector counts are averaged over many identical realizations.

Moreover, we can classify the jumps according to whether they come from trajectories where 1, 2 or 3+ photons are emitted into the waveguide (as indicated by the different bar colors in Fig. 5a). As we see in the plot, the higher the number detected in the waveguide, the earlier in time the jumps happen, in agreement with the simple theoretical model and with the calculations of $I_{out}(t)$ and $I_{out}^{(2)}(t, t)$, discussed above. We can also select only the trajectories where a single photon is detected at the waveguide output, and further separate those trajectories into two distinct cases: (i) when that is the only jump event (indicating a single photon was input and successfully propagated through the system), and (ii) where a multi-photon state was input, and all but one photon decayed into other channels. The histogram according to this classification in time is shown in Fig. 5b, where we see that the tail of faster arriving single photons, seen to the left of the main peak, results from the decay of number states with two of more photons, and the resulting mixing of propagation velocities.

Alternatively, we can use the jump statistics from a coherent state input to identify the intensity resulting from a Fock state input. Since the VIT system does not support any long lived excitations (compared with the simulated time scale), the total number of photon jumps (into any channel) out of the system for any one trajectory is equal to the number of the photons that entered the system for that trajectory. By post-selection on the total number of jumps we can then find the intensity that results from a Fock state input as shown in Fig. 5c. Here we see the same effect of jumps as noted above but observed in a different way. In particular, while we categorized the trajectories in Fig. 5a, b by the number of photons that survive and are output, in Fig. 5c we classify them by the number that are input. For Fock state inputs of two or more photons, the output intensities show tails of longer than expected delay times, again as a result of photon loss and the mixing of propagation speeds.

These altered delay times are not only due to quantum jumps however, they can also result from distortion of the multi-photon wavepacket as it enters the medium[30]. This distortion happens as the input pulse crosses the boundary of the atomic ensemble, as we illustrate for a two-photon wave function in Fig. 6a. For example, if the two photon wave function has a Gaussian pulse shape, the two photons can arrive at the boundary of the atomic ensemble at different times. The first photon that enters then travels at $v_1$ until the time that the second photon enters and the group velocity becomes $2v_1$. A similar process occurs when the photons exit the medium. In this case the further the photons are separated in the original pulse, the larger the delay of the photons. This process distorts the two-photon input Gaussian into a heart shaped output and higher photon number manifolds into higher dimensional hearts. In Fig. 6b we show how this behavior can be observed in the two time correlation measurement of the output photons for an input coherent pulse at low input photon number. For higher photon number input the heart shape is distorted as higher photon number manifolds with larger group velocity smear out the distribution.

## Discussion

In summary, we have described a novel technique to numerically simulate quasi-1D quantum light propagation through atomic ensembles, which is based on the powerful toolbox of matrix product states. This technique is versatile and adaptable to many cases of theoretical and experimental interest (e.g., with regard to

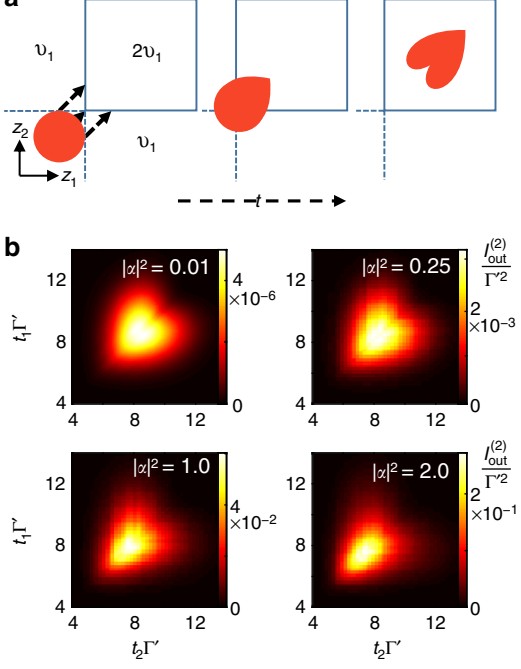

**Fig. 6** VIT pulse distortion. **a** Distortion process as a two-photon wave-function $\psi(z_1, z_2)$ enters the atomic medium. The initial Gaussian distribution of the photon positions $z_1$ and $z_2$ is shown as a circle. The two-dimensional space of the photon pair is divided into regions where only one photon is inside the medium and has group velocity $v_1$, indicated by the dashed lines, and when both photons are inside the medium having velocity $v_2 = 2v_1$, the square box. Photon pairs with greater separation spend more time in the regions where only one photon is inside the medium, delaying them compared to pairs with $z_1 = z_2$, leading to a characteristic heart shaped pattern. **b** Two-time correlation function $I_{out}^{(2)}(t_1, t_2)$ for the output field of the VIT system, after excitation with a coherent Gaussian input pulse for various average input photon number: $|\alpha|^2 = 0.01$, 0.25, 1.0, and 2.0. At weak input field the correlation function is purely due to the two photon component and shows a clean heart shape. As the number of input photons increases, higher photon number components contribute, which travel faster through the medium distorting the pattern and pulling it forward in time. The system parameters were for an optical depth of OD = 60, with $N = 30$, $\Gamma_{1D} = \Gamma'$, $\Delta = \delta_c = 0$, $g = 4$, $\Gamma'$, $\kappa = 0.03\Gamma'$, $\sigma_t = 4/\Gamma'$ and $t_0 = 6/\Gamma'$. We used bond dimension $D = 30$ and time step $\delta t = 0.01/\Gamma'$ where convergence was observed for all observables of interest (see Methods)

level structure, types of interactions, additional degrees of freedom, etc.). Similar to the important role that DMRG and MPS played in one-dimensional condensed matter systems, we envision that results gained from our numerical technique could be used to push forward the development of effective theories of strongly interacting systems of light[11, 12, 19, 22–25], and conversely that such analytical work could be used to improve numerical algorithms. Beyond that, it would be also interesting to investigate further why MPS apparently works well in the context of our open, long-range interacting system, and under what conditions MPS might fail. This could yield a better understanding of the growth of entanglement, which naively seems like a potentially useful resource, but which has not been explored for such systems to our knowledge.

The ability to formally map atom-light interactions to quantum spin models is intriguing in general, and it would be valuable to explore whether other techniques for solving spin systems give further insights into atom-light interactions. Finally, it should be

noted that this mapping essentially relies on the fact that the atomic response dominates the dispersion of near-resonant light fields, as compared to the dispersion of empty space. It would thus be interesting to investigate whether a similar effective theory could be derived for other strongly dispersive systems, such as exciton-polariton condensates[74–76], to shed new light on interacting photon dynamics in those settings.

## Methods

**Quantum jump formalism.** To find the time dynamics for the spin model we must evolve the master equation in time. Numerically this can be done directly by evolving the full density matrix $\rho$ in time using standard techniques such as the Runge-Kutta algorithm. Alternatively, we can instead use the "quantum jump" approach to unravel the master equation into trajectories of evolving pure states[66, 68]. Here we briefly review the quantum jump formalism, which we implement with MPS as discussed below.

We write the master equation for our 1D spin model in the form $\dot{\rho} = -i(H_{eff}\rho - \rho H_{eff}^\dagger) + \sum_l O_l\rho O_l^\dagger$, where $O_l$ are the "jump" operators associated with the dissipation resulting from emission into the waveguide and into free space, and $H_{eff}$ is a non-Hermitian effective Hamiltonian. This division of the master equation into jump terms and an effective Hamiltonian $H_{eff}$ is not unique and we attempt to do so here in a way that the jump operators have a physical significance. In particular, the emission of a photon into the forward going mode of the waveguide may interfere with the input light that is also traveling in the positive $z$ direction (Eq. (3)), an interference that would be present in real detection of photons output from the waveguide. This interference can be taken into account in our jump operator, and as such we take the forward going jump operator to be $O_+ = \mathcal{E}_{in}(t) + i\sqrt{\Gamma_{1D}/2}\sum_j e^{-ik_0 z_j}\sigma_{ge}^j$ (in contrast with $O_+ = \sqrt{\Gamma_{1D}/2}\sum_j e^{-ik_0 z_j}\sigma_{ge}^j$ as in ref. [36]). The backward going jump operator is simpler given the lack of input field in that mode, $O_- = i\sqrt{\Gamma_{1D}/2}\sum_j e^{ik_0 z_j}\sigma_{ge}^j$. In addition, we have $N$ local jump operators $O_j = \sqrt{\Gamma'}\sigma_{ge}^j$ corresponding to the free space decay, giving a set of possible jumps $O_l \in \{O_+, O_-, O_1, ..., O_N\}$.

With the jumps formulated in this way the effective Hamiltonian becomes

$$H_{eff} = H_0 - i\frac{\Gamma_{1D}}{2}\sum_{j,l=1}^{N}\exp(ik_0|z_j - z_l|)\sigma_{eg}^j\sigma_{ge}^l$$
$$-\sqrt{\frac{\Gamma_{1D}}{2}}\mathcal{E}_{in}(t)\sum_{j=1}^{N}e^{ik_0 z_j}\sigma_{eg}^j - \frac{i}{2}|\mathcal{E}_{in}(t)|^2. \quad (8)$$

In general $H_0$ can describe any additional atomic evolution; in the specific case of two level atoms coupled to a probe of frequency $\omega_P$ we can write, in the frame rotating with in the input frequency, $H_0 = -\sum_{j=1}^{N}(\Delta + i\Gamma'/2)\sigma_{ee}^j$, where $\Delta = \omega_P - \omega_{eg}$.

The quantum jump approach uses the above decomposition of the master equation to restate the evolution of the density operator as a sum of pure state evolutions called trajectories[66], where the wave function evolution is divided into (a) deterministic evolution under $H_{eff}$ and (b) stochastic quantum jumps made by applying jump operators $O_l$. Starting from a pure state $|\psi(t)\rangle$ at time $t$, the deterministic evolution over a time step $\delta t$ gives $|\psi(t + \delta t)\rangle = e^{-iH_{eff}\delta t}|\psi(t)\rangle$. However, during this evolution the norm of the state decreases to $\delta p = 1 - \langle\psi(t)|e^{iH_{eff}^\dagger\delta t}e^{-iH_{eff}\delta t}|\psi(t)\rangle$, as the effect of the jump operators is neglected. The effect of these operators is instead accounted for stochastically, where after each deterministic evolution we generate a random number $r$ between 0 and 1. If $r > \delta p$ the system remains in state $|\psi(t + \delta t)\rangle$. Otherwise, the state makes a random quantum jump to $|\psi(t + \delta t)\rangle = O_l|\psi(t)\rangle$ with probability $\delta p_l = \delta t\langle\psi(t)|O_l^\dagger O_l|\psi(t)\rangle$. The state is then normalized and the process repeats for the next time step and each sequence of evolutions gives a quantum trajectory. Any observable can be obtained by averaging its value over many trajectories. Furthermore, as we choose our quantum jumps to relate to physical processes, the distribution of the jumps can be thought of as corresponding to actual photon detection in an experiment.

**Time evolution with MPS.** As discussed in the Results section and depicted in Fig. 2, to evolve the MPS of the spin model in time and measure the expectation value of different observables we perform the following four steps.

(a)   Deterministic time evolution. To evolve the state $|\psi(t)\rangle$ in time we need to apply the operator $e^{-iH_{eff}\delta t}$ to the MPS representation. This is achieved by applying a matrix product operator (MPO) to the state, where just as a state can be decomposed into an MPS, any operator $W$ can be expressed in a local representation as

$$W = \sum_{\sigma'_1, ..., \sigma'_N, \sigma_1, ..., \sigma_N} W^{\sigma'_1, \sigma_1}W^{\sigma'_2, \sigma_2}...W^{\sigma'_N, \sigma_N}$$
$$\times |\sigma'_1, \sigma'_2, ..., \sigma'_N\rangle\langle\sigma_1, \sigma_2, ..., \sigma_N|. \quad (9)$$

Here $W^{\sigma'_j, \sigma_j}$ are a set of matrices at site $j$, where the matrices now have two physical indices $\sigma'_j, \sigma_j$ due to $W$ being an operator. An MPO may be "applied" to an MPS via a tensor contraction over the physical indices $\sigma_j$ of the MPS and MPO, as shown in Fig. 2a. This generates a new MPS with higher bond dimension, as the bond dimension of the MPO, $D_W$, multiplies the bond dimension of the original MPS, and for the calculation to be tractable $D_W$ must be small. Such a compact form is not known for the operator $e^{-iH_{\mathrm{eff}}\delta t}$; however, the first order approximation $e^{-iH_{\mathrm{eff}}\delta t} \approx 1 - iH_{\mathrm{eff}}\delta t$ has a compact MPO form if $H_{\mathrm{eff}}$ does. This is the case for the 1D spin model where the MPO representation of the effective Hamiltonian has $D_W = 4$. We can write $H_{\mathrm{eff}} = W_1 \ldots W_N$, where $W_j = \sum_{\sigma'_j, \sigma_j} W^{\sigma'_j, \sigma_j} \left| \sigma'_j \right\rangle \left\langle \sigma_j \right|$ are matrices of operators given by

$$W_j = \begin{pmatrix} \mathcal{I}^j & -\frac{i\lambda\Gamma_{1D}}{2}\sigma^j_{eg} & -\frac{i\lambda\Gamma_{1D}}{2}\sigma^j_{ge} & H^j_{loc} \\ 0 & \lambda\mathcal{I}^j & 0 & \sigma^j_{ge} \\ 0 & 0 & \lambda\mathcal{I}^j & \sigma^j_{eg} \\ 0 & 0 & 0 & \mathcal{I}^j \end{pmatrix}, \qquad (10)$$

for $1 < j < N$, and end vectors

$$W_1 = \begin{pmatrix} \mathcal{I}^1 & -\frac{i\lambda\Gamma_{1D}}{2}\sigma^1_{eg} & -\frac{i\lambda\Gamma_{1D}}{2}\sigma^1_{ge} & H^1_{loc} \end{pmatrix} \qquad (11)$$

and

$$W_N = \begin{pmatrix} H^N_{loc} & \sigma^N_{ge} & \sigma^N_{eg} & \mathcal{I}^N \end{pmatrix}^T. \qquad (12)$$

Here $\lambda = e^{ik_0 a}$, $\mathcal{I}^j$ is the identity operator for site $j$ and the $H^j_{loc}$ contain all the local terms in $H_{\mathrm{eff}}$. The MPO of the linear expansion of the time evolution operator $1 - iH_{\mathrm{eff}}\delta t$ can be obtained from this MPO without increasing the bond dimension. It is enough to replace $W_1$ with

$$W_1^{\mathrm{t.e.}} = \begin{pmatrix} -i\delta t\mathcal{I}^1 & -\frac{\delta t\lambda\Gamma_{1D}}{2}\sigma^1_{eg} & -\frac{\delta t\lambda\Gamma_{1D}}{2}\sigma^1_{ge} & \mathcal{I}^1 - i\delta tH^1_{loc} \end{pmatrix}, \qquad (13)$$

to obtain the desired MPO. Using a small time step $\delta t$ we can then advance the wave function in time by applying this MPO.

(b) Quantum jumps. After evolving a time $\delta t$, the state is either kept and renormalized, or a jump is applied. To apply the quantum jump formalism we then just require an MPO form of the jump operators that can be applied to the MPS at each time step, see Fig. 2b. The jump operators of the 1D spin model can be written in compact MPO form, where loss into the waveguide requires an MPO of bond dimension $D_W = 2$. The jump operator corresponding to the emission of a photon in the $+z$ output channel can be written as $O_+ = Z_1 Z_2 \ldots Z_N$ with

$$Z_j = \begin{pmatrix} \mathcal{I}^j & i\sqrt{\frac{\Gamma_{1D}}{2}}e^{-ik_0 z_j}\sigma^j_{ge} \\ 0 & \mathcal{I}^j \end{pmatrix}, \qquad (14)$$

for $1 < j < N$, and end vectors

$$Z_1 = \begin{pmatrix} \mathcal{I}^1 & \mathcal{E}(t)\mathcal{I}^1 + i\sqrt{\frac{\Gamma_{1D}}{2}}e^{-ik_0 z_1}\sigma^1_{ge} \end{pmatrix} \qquad (15)$$

and

$$Z_N = \begin{pmatrix} i\sqrt{\frac{\Gamma_{1D}}{2}}e^{-ik_0 z_N}\sigma^N_{ge} & \mathcal{I}^N \end{pmatrix}^T. \qquad (16)$$

The MPO of $O_-$ is analogous, but without the external field term in $Z_1$ and with $k_0$ replaced by $-k_0$. For the jumps associated with spontaneous emission into free space an MPO representation is not required as these jumps just require an operator to be applied locally to a single site.

(c) State compression. After applying the time evolution operator or jump operators the size of the MPS increases as the bond dimension of the operator multiplies the bond dimension of the original state. Over time this would lead to exponential growth in the MPS size if not constrained. This increase in bond dimension can correspond to the true build up of entanglement, but may also correspond to the new state being expressed inefficiently in the MPS form. In the second case, a more efficient representation can be found and the bond dimension compressed to a smaller value, as in Fig. 2c. This can be done using singular value decompositions to find low rank approximations of the matrices $A^{\sigma_j}$ in the MPS representation, or by variationally exploring the space of MPS states with a fixed bond dimension that are closest to the original state[31, 32]. The validity of such a compression can be evaluated by checking how strongly the parts of the state discarded in the compression contribute to the description. From this an error can be calculated and the bond dimension in the compression adjusted so the error remains small (see below).

(d) Calculating observables. At any point in time observables such as the spin populations or output field may be calculated for a particular quantum

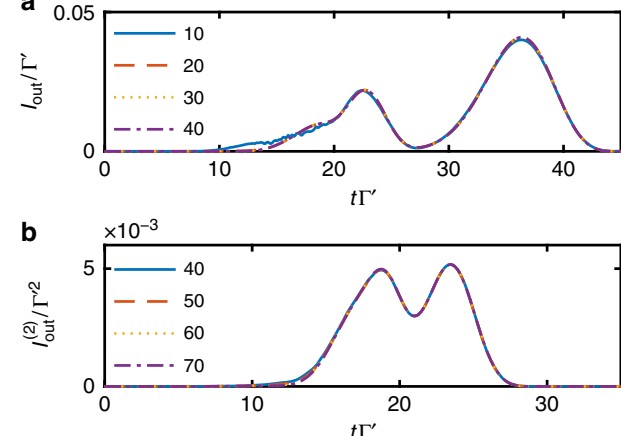

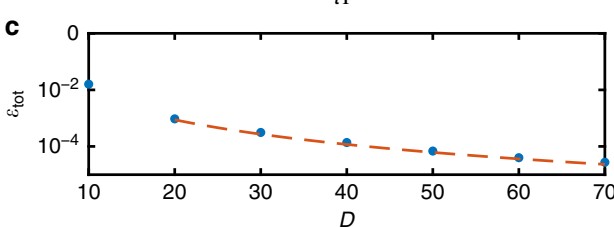

**Fig. 7** Convergence of VIT observables. **a** Intensity and **b** zero-delay second-order correlation function for the quantum trajectory without jumps at different values of the maximum bond dimension $D$. **c** Log-scale accumulated compression error $\varepsilon_{\mathrm{tot}}$ as function of the MPS maximum bond dimension $D$ (blue dots). The red line is a polynomial fit, $\varepsilon_{\mathrm{tot}} \propto D^{-2.9}$, for all the points except $D = 10$

trajectory by applying the appropriate operator associated with that observable in MPO form to the state. For example, to find the output intensity, $\langle \psi(t) | E^\dagger_{\mathrm{out}}(t) E_{\mathrm{out}}(t) | \psi(t) \rangle$, one can express the individual elements as matrix product states or operators. The intensity for that trajectory can then be evaluated through a tensor contraction, as shown in Fig. 2d. This intensity is then averaged over all the quantum trajectories to find the expectation value $I_{\mathrm{out}}(t) = \langle E^\dagger_{\mathrm{out}}(t) E_{\mathrm{out}}(t) \rangle$. Multi-time correlation functions such as $I^{(2)}_{\mathrm{out}}(t, t+\tau) = \langle E^\dagger_{\mathrm{out}}(t) E^\dagger_{\mathrm{out}}(t+\tau) E_{\mathrm{out}}(t+\tau) E_{\mathrm{out}}(t) \rangle$ can also be found. This is done by propagating the state in time until time $t$ and then applying the operator $E_{\mathrm{out}}$ to the state. The state is evolved a further time $\tau$ and the operator applied again. The norm of the resulting states are then averaged over many such evolutions to find the two-time correlation.

**VIT matrix product operators**. The MPO of the VIT Hamiltonian can be obtained by extending the bare spin model case above. The cavity degree of freedom is associated with an additional site in the spin chain at position $N+1$, in which case the VIT MPO of $H_{\mathrm{eff}}$ is obtained by adding two columns and rows to the bare representation:

$$W_j^{\mathrm{VIT}} = \begin{pmatrix} \cdots & \cdots & \cdots & \frac{g}{2}\sigma^j_{es} & \frac{g}{2}\sigma^j_{se} & \cdots \\ \cdots & \cdots & \cdots & 0 & 0 & \cdots \\ \cdots & \cdots & \cdots & 0 & 0 & \cdots \\ 0 & 0 & 0 & \mathcal{I}^j & 0 & 0 \\ 0 & 0 & 0 & 0 & \mathcal{I}^j & 0 \\ \cdots & \cdots & \cdots & 0 & 0 & \cdots \end{pmatrix}, \qquad (17)$$

for $1 < j \leq N$, where the dots stand for the elements given in Eq. (10) and

$$W_{N+1}^{\mathrm{VIT}} = \begin{pmatrix} H_{\mathrm{loc,cav}} & 0 & 0 & b & b^\dagger & \mathcal{I}_{\mathrm{cav}} \end{pmatrix}^T. \qquad (18)$$

**Convergence and accumulated error**. Evolving the MPS representation of a state through time increases the bond dimension of the MPS. In particular, the action of the time evolution MPO, with maximum bond dimension $D_W$, on an MPS, with maximum bond dimension $D$, increases the bond dimension to $D' = D_W \times D$. In our simulations we typically keep the maximum bond dimension $D$ of the MPS fixed throughout the evolution and to do so it is necessary to compress the MPS from dimension $D'$ to $D$ after each step. This allows for efficient computation,

however, for the results of the simulation to match reality, this compression must be done in a controlled manner to avoid discarding important information from the state.

One straightforward way to check the validity of the simulations is then to do the same simulation for various maximum bond dimensions $D$ and see that values of the observables of interest converge as $D$ increases. In Fig. 7a, b we plot the output intensity and zero-delay second-order correlation function for different bond dimensions for the trajectory without jumps (similar results hold for the other trajectories) for VIT with parameters as given in Fig. 4. We see that the intensity has an excellent convergence already for $D = 20$, while the zero-delay second-order correlation function requires higher bond dimension $D\sim50$ for convergence. From this we conclude that to accurately model the higher number components of the pulse requires larger bond dimension, as the higher number components of the pulse have higher weight in the second order correlation function. Correspondingly, in our simulations we fix the bond dimension to $D = 50$.

Another way to ensure the simulations accurately model the physical reality is to monitor the error incurred in each compression step. The compression can be done variationally minimizing the distance between the larger MPS $|\psi(t)\rangle_{D'}$ and its compressed version $|\psi(t)\rangle_D$, or by a sequence of singular value decompositions of the bond connections between each site[32]. In the latter case, the large bond dimension $D'$ yields $D'$ singular values $\lambda_{t,j,l}$ at bond site $j$ and time $t$, with $1 \leq l \leq D'$. If we order the singular values to monotonically decrease with increasing $l$, we may then reduce the bond dimension by keeping only the singular values with $l \leq D$. One measure of this compression error is the norm of the difference of the original state and the compressed state $\epsilon_t = \left\||\psi(t)\rangle_D - |\psi(t)\rangle_{D'}\right\|$, which can be expressed in terms of the discarded singular values, where $\epsilon_t = 1 - \prod_{j=1}^{N-1}\left(1 - \epsilon_{t,j}\right)$ with $\epsilon_{t,j} = \sum_{l>D}\lambda_{t,j,l}^2$. The error accumulated during the whole time evolution is $\epsilon_{\text{tot}} = 1 - \prod_t (1 - \epsilon_t)$. Since all the terms are small one can approximate the products with sums and obtain

$$\epsilon_{\text{tot}} \approx \sum_{t=0}^{T_f} \sum_{j=1}^{N-1} \sum_{l>D} \lambda_{t,j,l}^2, \qquad (19)$$

giving a figure of merit for the overall quality of the time evolution. In Fig. 7c, we plot the accumulated error for different bond dimensions. We find that starting from $D = 20$ the accumulated error decays as a power law $\epsilon_{\text{tot}} \sim D^{-\alpha}$, with $\alpha \approx 2.9$.

**Data and code availability**. The data presented in the figures of this manuscript are available from the corresponding author upon request. The code used for the MPS simulation of the spin model is available at https://github.com/jdnz/MatrixProductStates.jl.

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

## Acknowledgements

We thank J.J. Garcia-Ripoll and L. Mathey for stimulating discussions. This work was supported by Fundacio Privada Cellex Barcelona, the CERCA Programme/Generalitat de Catalunya, the MINECO Ramon y Cajal Program, the Spanish Ministry of Economy and Competitiveness, through the Severo Ochoa Programme for Centers of Excellence in R&D (SEV-2015-0522) and Plan Nacional Grant CANS, the Marie Curie Career Integration Grant ATOMNANO, the ERC Starting Grant FoQAL, the European Commission FET Open XTrack Project GRASP, the US MURI Grants QOMAND and Photonic Quantum Matter, and la Caixa-Severo Ochoa PhD Fellowship.

## Author contributions

J.S.D. and M.T.M. wrote the MPS code and performed the calculations. All authors contributed ideas and to writing the manuscript.

## Additional information

**Competing interests:** The authors declare no competing financial interests.

