## [Peer Review File · Nature Communications]

Reviewers' comments:

Reviewer #1 (Remarks to the Author):

report on manuscript NCOMMS-17-04153

„Simulating quantum light propagation through atomic ensembles using matrix product states“

by M.T. Manzoni, D. Chang, and J.S. Douglas

The present paper introduces an interesting approach for a numerical treatment of quantum many-body effects of light pulses interacting with atomic ensembles in one spatial dimension. This is a challenging problem in particular in the regime of strong nonlinear effects and in the presence of decay.

At the same time experimental techniques in particular those employing the coupling of light to ensemble of Rydberg atoms have developed to the point that many-body effects in light propagation can be observed in the strong interaction regime.

Thus there is clear need for numerical methods.

Standard approaches are limited to few-particle wave function simulations or approximative methods. In the present paper the authors suggest to map the light propagation problem to one of interacting spins with long-range interactions. To this end the radiation field is formally integrated out in terms of Greensfunctions of the electromagnetic field. The dynamics of the spin model is then treated using a matrix-product representation of the state vector combined with a wave-function Monte-Carlo

approach to treat dissipation. To illustrate their method the authors study in particular light propagation in vacuum induced transparency, for which interesting nonlinear effects such as photon-number dependent group delay have been predicted.

I believe the manuscript merits publication but it does have a number of shortcomings which need to be addressed before I could recommend publication.

(i) In contrast to MPS based ground-state methods, the simulation of time-dependent problems using eg. TEBD suffers from the often linear-in-time growth of entanglement entropy. This limits the applicability of TEBD methods to the short time regime. I am missing a more detailed discussion of this issue in the paper. What is the dynamics of the entanglement entropy in the example studied and under what general conditions can one expect that it stays low enough for the method to be applicable to interesting problems?

(ii) In many experimentally relevant problems of strong atom-light interaction, e.g. in the interaction with Rydberg atoms, a large number of atoms are needed. TEBD simulations can typically handle up to a hundred spins. To what extend can the method of the authors be applied to problems other than those with strong (radial) light confinement (such that beam cross section is on the order of λ^2), where strong effects can be seen already for few atoms?

Some more specific comments:

(1) In the introduction the authors say: „In the problem of light propagation through atomic ensemble, we show that the only independent degree of freedom are the atomic internal states.“ Integrating out the field degrees of freedom to map the light-matter interaction to non-local (and in general non-instantaneous) spin interactions is not new and goes back to Schwingers source theory. Eqs.(6) and (7) for one-dimensional waveguides have been derived before in the literature. It should also be noted that the sinusoidal interaction eq.(6) is not correct at very short distances (see e.g. PRB 84, 075419 (2011)).

(2) In the discussion of effects of decay in VIT one should distinguish between effects coming from cavity decay and atomic absorption losses. As argued in Ref.[20] cavity decay has no effect on the separation of the individual photon components.

Also in the example chosen in Fig.4 is the pulse bandwidth smaller than the EIT bandwidth?

(3) The letter „a“ is used in section IV for different things (annihilation operator , length scale).

Reviewer #2 (Remarks to the Author):

The manuscript describes a very complete analysis of light propagation through an atomic ensemble using matrix product states (MPS). The paper is extremely well written and the results are sound and important for the scientific community. However, even though MPS is a numerically exact method, the model to be solved is based on two approximations that in my opinion are not sufficiently justified. In detail:

The authors should clarify in my opinion what is the origin or justification of considering the emission in the other channels (other than the 1D considered) in terms of an independent decay channel Γ' (as specified before Eq. (8)). In the 3D atomic ensemble case the EM field modes depend on the 3D wave vector, i.e. the harmonic annihilation operators are $a_{\mathbf{k}}$, and therefore it is not so straightforward to decompose the interaction Hamiltonian into 1D emission + 2D emission. Perhaps the authors could explain a bit more their approach, or at least give some reference where such approach is rigorously justified.

In addition to that, the analysis seem to depend on a Markov approximation. Could the authors provide any physical argumentation of why such an approximation should be accurate in the regime they are considered? In detail, the Markov approximation relies on assuming that the interaction with the atoms only produces a small fluctuation of the environment state around its equilibrium (in this case the vacuum state). I would expect such approximation to fail when considering strong collective effects (i.e. cooperative emission occurring at high optical densities) and many excitations in the problem. While I can understand that there might be conditions under which the Markov approximation might still be valid in all the regimes here considered, I would like the authors to clarify these or at least to provide a discussion on the subject.

I think these two points should be clarified by the authors, given the fact that the aim of the paper is not so much to work on a toy model, but mostly to account for a physically realistic situation with as much accuracy as possible.

Reviewer #3 (Remarks to the Author):

In this manuscript, the authors develop a novel framework for calculation of quantized light propagation through an (atomic) medium using matrix product states to attack problems which so far have been beyond reach of numerical (and analytical) approaches. The developed framework is then applied to the case of vacuum-induced transparency, a phenomena that has only recently been realized experimentally and which has been attacked by multiple theory approaches. The authors can show very convincingly the power of their method in the case of this example, which in itself will be of great interest to others working on this topic.

The paper is a highly interesting combination of state-of-the-art topics from quantum optics and

condensed matter physics and should thus be of interest to a wide range of readers. It is very well written and manages to convey its complicated topics from different fields very comprehensively. I have a few minor comments below, but apart from asking the authors to briefly consider these, I highly recommend publication of this paper in Nat. Comm.

Comments:

1) The introduction gives a good overview of where the authors place their work in relation to both experimental and theory efforts on few-photon propagation through optical media. There is of course a whole field "coming from the other side", i.e. semiconductor exciton-polaritons, where the interaction is weak and photon number is large. But similarly, this field borrows heavily from condensed-matter physics using well-developed theory for weakly interacting systems to study interacting photon fluids. It would be nice if the authors could briefly comment on how their method might bridge to this regime and if it could for example shed new light on the transition from weak (many photons) to strong (few photons) interaction.

2) To me, the main part of the paper is section III, where the novel method is very well presented, and section IV where it is applied to give a very thorough analysis of VIT. In contrast, I found section II rather lengthy and to some extent repeating "well-established" knowledge (at least to the quantum optics community). I personally suggest moving some parts from section II, mostly the quantum jump discussion, to the appendix. That is my personal opinion though, if the authors think the paper benefits from this broad overview, they should keep it as it is.

3) To me the argument about reducing the total number of atoms, but in turn increasing the single-atom coupling strength to reproduce the correct OD, not totally clear. How I read it, it is argued that one can even use a coupling to the waveguide mode significantly stronger than the scattering rate? That seems strange to me, but maybe I just misread what is written. And secondly, the number of atoms should be (much) larger than the number of photons to avoid nonlinearity due to saturation? Finally, I see why this in principle works when H_0 is a single atom Hamiltonian (as in the VIT example). In contrast, how well this approach does if H_0 includes e.g. atom-atom interaction seems far less obvious, because the atomic dynamics then may scale in some way with N (beyond simply OD scaling with N). I would ask the authors to extend the discussion of these points.

We thank the referees for their comments and for their suggestions on how to improve our manuscript. In response we have made significant revisions to take these suggestions into account, which we believe have considerably strengthened the manuscript.

Before responding to the comments/suggestions of the referees on a point-by-point basis below, we will first discuss a major change we made to the initial structure of the paper motivated by comments from all three reviewers. This change consists of removing the initial treatment of light propagation with the Green's function formalism and instead motivating our MPS model based on a direct mapping from the paraxial Maxwell-Bloch equations. With this new structure we aim to clarify the assumptions behind our model and at the same time simplify the treatment.

The paraxial Maxwell-Bloch equations, as now described in the first section of our manuscript, are the standard theoretical tools used to study light propagation in atomic ensembles, capturing physics such as photon storage in quantum memories and non-linear propagation in Rydberg gases. While these equations do not describe all phenomena associated with ensembles, for example trapping of radiation in the transversal direction, we can show that when they do apply, there exists a direct mapping to the equations of light propagation in a one-dimensional waveguide coupled to trapped atoms. This mapping then allows us to take advantage of the fact that light propagation in such a waveguide is typically not dependent on atom number explicitly, allowing us to reduce the atom number in our model by adjusting other parameters. The smaller number of atoms can then be treated directly using MPS providing a solution to original Maxwell-Bloch equations of the atomic ensemble.

In this way we hope to clarify that our model is valid under the same conditions as the paraxial Maxwell-Bloch equations. The subject of when such equations are valid is a subtle one and goes beyond the scope of our paper, however we have added new references [20, 50] where the map from 3D to 1D is studied systematically.

Below we make specific responses to the comments made by each reviewer.

Response to Reviewer #1

- 1) **Referee comment:** "In contrast to MPS based ground-state methods, the simulation of time-dependent problems using eg. TEBD suffers from the often linear-in-time growth of entanglement entropy. This limits the applicability of TEBD methods to the short time regime. I am missing a more detailed discussion of this issue in the paper. What is the dynamics of the entanglement entropy in the example studied and under what general conditions can one expect that it stays low enough for the method to be applicable to interesting problems?"

Reply: The scaling of entanglement in a general driven-dissipative system is currently unknown. However, as we discuss in the first paragraph of page 5 of the manuscript

there are a number of reasons why we believe MPS will be efficient in treating propagation problems. Specifically, as opposed to a closed system where entanglement can grow indefinitely with time, in photon propagation, photons are continuously leaving the system limiting the entanglement growth. There are then two possible cases: the input is a pulse and the number of photons in the pulse limits entanglement, or continuous driving where the state will eventually reach a steady state with fixed entanglement. In both cases the level of entanglement could still be beyond what is computationally possible, however in practice we are yet to find a situation where that is the case. For pulsed input, as studied in the VIT example, we have added a supplementary document describing how the entanglement entropy scales as the pulse propagates with arbitrary shape in the atomic medium. In this document, we also show that in the case of VIT entanglement entropy remains small.

- 2) **Comment:** “In many experimentally relevant problems of strong atom-light interaction, e.g. in the interaction with Rydberg atoms, a large number of atoms are needed. TEBD simulations can typically handle up to a hundred spins. To what extent can the method of the authors be applied to problems other than those with strong (radial) light confinement (such that beam cross section is on the order of λ^2), where strong effects can be seen already for few atoms?”

Reply: As discussed above, we have rewritten the manuscript to emphasize that our method applies to situations modeled by the paraxial Maxwell-Bloch equations. In this case, the physics of a large ensemble of atoms (e.g. 10^6 atoms) can be modeled by a few hundred atoms making the MPS treatment possible. It should be noted that strong radial confinement of the light is not required for the Maxwell Bloch equations to be a reasonable model, which then significantly extends the range of systems where our MPS treatment can be applied.

As one example, in ongoing work in collaboration with an experimental group, we have used the MPS method to model propagation at high photon flux through a cold atomic Rydberg ensemble, under conditions of electromagnetically induced transparency. There a Gaussian paraxial beam passes through tens of thousands of atoms and we have obtained good agreement using our model with only tens of atoms, by matching both the optical depth and the optical depth per blockade radius of the two systems. (We realize that this statement must be taken on faith as this ongoing work is still unpublished, but in any case we believe that our new discussion relating the Maxwell-Bloch equations and MPS should be convincing.)

- 3) **Comment:** “In the introduction the authors say: ‘In the problem of light propagation through atomic ensemble, we show that the only independent degree of freedom are the atomic internal states.’ Integrating out the field degrees of freedom to map the light-matter interaction to non-local (and in general non-instantaneous) spin interactions is not new and goes back to Schwingers source theory. Eqs. (6) and (7)

for one-dimensional waveguides have been derived before in the literature. It should also be noted that the sinusoidal interaction eq.(6) is not correct at very short distances (see e.g. PRB 84, 075419 (2011)).”

Reply: We agree that the idea of integrating out the photonic degrees of freedom is not new. Certainly in general settings, the dipole-dipole interactions that result have been discussed and utilized in many different contexts historically, ranging from the classic Gross and Haroche review on superradiance (Phys. Rep., 1982) to Kurizki’s investigation of dipole-dipole interactions in photonic crystals in the 1990’s to Welsch and Buhmann’s quantization of electromagnetic fields in dielectrics in the 1990’s and early 2000’s. We have added a representative sample of references to the text on page 2: “The essence of the spin model is to recognize that in light propagation through an ensemble, the only independent degrees of freedom are the atomic internal states (“spins”). In particular, the light fields mediate interactions between the atoms and a common theoretical approach is to integrate out the fields, reducing the description to a problem of N interacting spins (N being the number of atoms) [33-35].”

In addition, we completely agree that for a given physical system such as actual atoms coupled to a nanophotonic waveguide, other terms in the atomic dipole-dipole interaction besides a sinusoidal interaction can be present due to a number of reasons (e.g., contributions from free-space modes). However, the primary goal of our work is not to capture such system-specific details of an atom-nanophotonics interface. Rather, and as made more clear by our revisions, our goal is to capture dynamics equivalent to the Maxwell-Bloch equations. The “toy model” for the 1D waveguide that we have presented exactly encodes the Maxwell-Bloch equations, and without additional terms that could appear due to a more system-specific Green’s function. To clarify that in actual 1D systems, such as nano-wires, there are additional effects when the atoms are spaced with distances much less than the waveguide we added the following sentence to the end of the final paragraph of the 1D spin model section: “In these cases, our model is valid when the spacing between the atoms is of the order of the wavelength of the light or larger, for smaller atomic distances additional effects can occur [51,64]”.

- 4) **Comment:** “In the discussion of effects of decay in VIT one should distinguish between effects coming from cavity decay and atomic absorption losses. As argued in Ref.[20] cavity decay has no effect on the separation of the individual photon components. Also in the example chosen in Fig.4 is the pulse bandwidth smaller than the EIT bandwidth?”

Reply: In our simulation results, we can separate the propagation effects resulting from cavity decay and spontaneous emission from the excited state. However, in general we find that the two processes have a qualitatively similar effect on propagation, where both decays effect the separation of the photon components. For example, during propagation a two-photon state can decay via cavity decay (or spontaneous emission)

leaving only a single photon in the forward propagating mode. The propagation velocity then changes from the two-photon velocity v_2 to the one-photon velocity v_1 . (We note that the remaining photon still propagates as a dark-state polariton, which may be where the point of confusion arises.) The mix of these two velocities then leads to single photons arriving at the output significantly earlier than if a single photon propagated by itself. In the extreme case where this decay happens just before the two photons would have exited the medium, a single photon is detected with apparent delay time L/v_2 instead of L/v_1 . Depending on the strength of the decay rates, this effect can destroy the separation between detected single photons and bi-photons as we see in our simulations.

In the figure below we plot data from trajectories where there was only a single photon detected in the output channel, the forward-going waveguide mode, given a coherent state input pulse. The curves show the time-binned number of counts detected in this mode for three different situations. This blue curve represents single photons detected when no other jumps into other channels occurred, i.e., single photon in, single photon out. The other curves are single photons detected from trajectories in which jumps occurred via cavity decay only (green) and spontaneous decay only (red), i.e., the single photon output resulted from a decayed higher photon number state. Here we see that cavity decay alone and spontaneous decay alone both lead to single photons arriving earlier than expected if this loss was ignored.

The photon pulse we used in Fig. 4 had temporal field envelope $\exp(-t^2/\sigma_t^2)$ with $\sigma_t = 3/\Gamma'$. In this case the pulse bandwidth $\sim 2/\sigma_t = 2\Gamma'/3$ is compatible with the EIT bandwidth $\sim g^2/\Gamma'/\sqrt{OD} = 4\Gamma'/5$ for the parameters used. In practice, we chose the pulse width numerically in an effort to emphasize the pulse separation, while at the same time not introducing significant distortion and loss.

- 5) **Comment:** “The letter „a“ is used in section IV for different things (annihilation operator, length scale).”

Reply: We have replaced the operator a with b .

Response to Reviewer #2

- 1) **Comment:** “The authors should clarify in my opinion what is the origin or justification of considering the emission in the other channels (other than the 1D considered) in terms of an independent decay channel Γ' (as specified before Eq. (8)). In the 3D atomic ensemble case the EM field modes depend on the 3D wave vector, i.e. the harmonic annihilation operators are $a_{\mathbf{k}}$, and therefore it is not so straightforward to decompose the interaction Hamiltonian into 1D emission + 2D emission. Perhaps the authors could explain a bit more their approach, or at least give some reference where such approach is rigorously justified.”

Reply: As discussed above, we have changed the manuscript to emphasize that our model applies in the same regime as the paraxial Maxwell-Bloch equations, which typically assume that decay out of the paraxial mode occurs independently. As such, our model does not describe cases where this assumption breaks down, such as radiation trapping or in ordered atomic arrays where the interatomic distance is smaller than the wavelength (see Ref. [51]). In the revised manuscript, we cite in particular Refs. [20, 50] where the map from 3D to 1D is studied systematically, and a set of other references where paraxial Maxwell-Bloch equations are employed for the study of light propagation in atomic ensembles.

- 2) **Comment:** “In addition to that, the analysis seem to depend on a Markov approximation. Could the authors provide any physical argumentation of why such an approximation should be accurate in the regime they are considered? In detail, the Markov approximation relies on assuming that the interaction with the atoms only produces a small fluctuation of the environment state around its equilibrium (in this case the vacuum state). I would expect such approximation to fail when considering strong collective effects (i.e. cooperative emission occurring at high optical densities) and many excitations in the problem. While I can understand that there might be conditions under which the Markov approximation might still be valid in all the regimes here considered, I would like the authors to clarify these or at least to provide a discussion on the subject.”

Reply: The reviewer is correct in that our treatment requires a form of Markov approximation and we have added text following equation (3) describing this. The particular consequence of the Markov approximation is that time retardation of the electric field is neglected in Eq. (3), so that the field depends on the atomic coherence at the same time. This approximation is valid when the delay L/c in propagating the

length of the system L is much smaller than the time scale characterising the atomic evolution, typically of the order of $1/\Gamma'$. A complementary viewpoint is that this assumes that the atoms are the dominant source of pulse dispersion. This is clearly the case in EIT or VIT, where the extremely slow group velocities associated with the dark-state polaritons only depend on the atomic properties, and not on the speed of light in empty space (note that this treatment neglects propagation delay for the free electric field, but not delay from coupling to the atomic response). Aside from that approximation, however, collective effects and multiple excitation effects are fully included in our model as the many-body spin state encodes the full structure of the electromagnetic field (although it may be that a cooperative effect like superradiance shortens the time scale characterizing the atomic evolution and makes more stringent the conditions for validity of the model).

Response to Reviewer #3

- 1) **Comment:** “The introduction gives a good overview of where the authors place their work in relation to both experimental and theory efforts on few-photon propagation through optical media. There is of course a whole field “coming from the other side”, i.e. semiconductor exciton-polaritons, where the interaction is weak and photon number is large. But similarly, this field borrows heavily from condensed-matter physics using well-developed theory for weakly interacting systems to study interacting photon fluids. It would be nice if the authors could briefly comment on how their method might bridge to this regime and if it could for example shed new light on the transition from weak (many photons) to strong (few photons) interaction.”

Reply: That is a very interesting question raised by the referee. As described further in our reply to comment 2) of reviewer #2, our spin model essentially relies on the fact that atoms near resonance yield much more dispersion to an electromagnetic field than empty space (or more generally beyond atoms, that the material polarization energy dominates the free field energy within the material). This in turn enables the field to be effectively eliminated. Certainly a system such as an exciton-polariton condensate seems to exhibit the same essential ingredient. However, the specific route toward generalizing to a “real material” does not seem straightforward, and likely requires a careful derivation far beyond the scope of the current paper. In which case we have simply posed this as an open question in the outlook: “Finally, it should be noted that this mapping essentially relies on the fact that the atomic response dominates the dispersion of near-resonant light fields, as compared to the dispersion of empty space. It would thus be interesting to investigate whether a similar effective theory could be derived for other strongly dispersive systems, such as exciton-polariton condensates [74{76}], to shed new light on interacting photon dynamics in those settings.”

- 2) **Comment:** “To me, the main part of the paper is section III, where the novel method is very well presented, and section IV where it is applied to give a very thorough analysis

of VIT. In contrast, I found section II rather lengthy and to some extent repeating "well-established" knowledge (at least to the quantum optics community). I personally suggest moving some parts from section II, mostly the quantum jump discussion, to the appendix. That is my personal opinion though, if the authors think the paper benefits from this broad overview, they should keep it as it is."

Reply: We have followed the Referee's suggestion and we have moved the discussion of the quantum jump formalism into the Methods.

- 3) **Comment:** "To me the argument about reducing the total number of atoms, but in turn increasing the single-atom coupling strength to reproduce the correct OD, not totally clear. How I read it, it is argued that one can even use a coupling to the waveguide mode significantly stronger than the scattering rate? That seems strange to me, but maybe I just misread what is written. And secondly, the number of atoms should be (much) larger than the number of photons to avoid nonlinearity due to saturation? Finally, I see why this in principle works when H_0 is a single atom Hamiltonian (as in the VIT example). In contrast, how well this approach does if H_0 includes e.g. atom-atom interaction seems far less obvious, because the atomic dynamics then may scale in some way with N (beyond simply OD scaling with N). I would ask the authors to extend the discussion of these points."

Reply: As we discuss in the manuscript, provided we choose the atomic spacing of the atoms in the 1D chain correctly, the 1D model reproduces light absorption in an atomic ensemble with optical depth, $OD = 2 \Gamma_{1D} N / \Gamma'$ in our notation. We can then reproduce the same optical depth using less atoms but higher Γ_{1D} . However, as the reviewer points out, this breaks down if Γ_{1D} becomes too large compared to Γ' , or if atoms begin to saturate. In practice, we find for VIT that we can choose $\Gamma_{1D} \sim \Gamma'$ without saturation effects appearing. Moreover, the validity of reducing the atom number can be checked by increasing the atom number while keeping the same optical depth and checking for numerical convergence.

For the case of interactions between the atoms, the number of atoms may also not appear explicitly in the propagation problem, where typically the important quantity is the optical depth within the interaction range. For instance, in the case of Rydberg atoms, the relevant quantity for light propagation is the optical depth per Rydberg blockade. This optical depth can then be treated in the same way as the full optical depth above, and again the independence of the system from N alone can be numerically verified. We have added text discussing the above points on page 4 of the manuscript. Finally, in some hypothetical case where the photon dynamics do explicitly depend on N , such a dependence would be revealed directly in the numerical scaling checks, which would itself give rise to important physical insight.

REVIEWERS' COMMENTS:

Reviewer #1 (Remarks to Author):

I have read the revised version of the manuscript and the authors detailed reply. The structural change at the beginning of the paper has clearly improved the work and has made it more accessible to the general readership. I am pretty much happy with the manuscript except concerning my first comment. I believe however that this can be taken care of easily without major revision and the manuscript should be published.

Original comment: In many experimentally relevant problems of strong atom-light interaction, e.g. in the interaction with Rydberg atoms, a large number of atoms are needed. TEBD simulations can typically handle up to a hundred spins. To what extent can the method of the authors be applied to problems other than those with strong (radial) light confinement (such that beam cross section is on the order of λ^2), where strong effects can be seen already for few atoms?

Reply: As discussed above, we have rewritten the manuscript to emphasize that our method applies to situations modeled by the paraxial Maxwell-Bloch equations. In this case, the physics of a large ensemble of atoms (e.g. 10^6 atoms) can be modeled by a few hundred atoms making the MPS treatment possible. It should be noted that strong radial confinement of the light is not required for the Maxwell Bloch equations to be a reasonable model, which then significantly extends the range of systems where our MPS treatment can be applied.

As one example, in ongoing work in collaboration with an experimental group, we have used the MPS method to model propagation at high photon flux through a cold atomic Rydberg ensemble, under conditions of electromagnetically induced transparency. There a Gaussian paraxial beam passes through tens of thousands of atoms and we have obtained good agreement using our model with only tens of atoms, by matching both the optical depth and the optical depth per blockade radius of the two systems. (We realize that this statement must be taken on faith as this ongoing work is still unpublished, but in any case we believe that our new discussion relating the Maxwell-Bloch equations and MPS should be convincing.)

I am satisfied with the authors reply. It seems reasonable that for effective one-dimensional models the number of spins can be sufficiently large for simulations. I also trust the authors that they have checked this for a specific application using Rydberg ensemble

Original comment: In the introduction the authors say: In the problem of light propagation through atomic ensemble, we show that the only independent degree of freedom are the atomic internal states. Integrating out the field degrees of freedom to map the light-matter interaction to non-local (and in general non-instantaneous) spin interactions is not new and goes back to Schwingers source theory. Eqs. (6) and (7) for one-dimensional waveguides have been derived before in the literature. It should also be noted that the sinusoidal interaction eq.(6) is not correct at very short distances (see e.g. PRB 84, 075419 (2011)).

Reply: We agree that the idea of integrating out the photonic degrees of freedom is not new. Certainly in general settings, the dipole-dipole interactions that result have been discussed and utilized in many different contexts historically, ranging from the classic Gross and Haroche review on

superradiance (Phys. Rep., 1982) to Kurizki's investigation of dipole-dipole interactions in photonic crystals in the 1990's to Welsch and Buhmann's quantization of electromagnetic fields in dielectrics in the 1990's and early 2000's. We have added a representative sample of references to the text on page 2: "The essence of the spin model is to recognize that in light propagation through an ensemble, the only independent degrees of freedom are the atomic internal states ("spins"). In particular, the light fields mediate interactions between the atoms and a common theoretical approach is to integrate out the fields, reducing the description to a problem of N interacting spins (N being the number of atoms) [33-35]."

In addition, we completely agree that for a given physical system such as actual atoms coupled to a nanophotonic waveguide, other terms in the atomic dipole-dipole interaction besides a sinusoidal interaction can be present

The revised text is now much clearer on this.

Original comment: In the discussion of effects of decay in VIT one should distinguish between effects coming from cavity decay and atomic absorption losses. As argued in Ref.[20] cavity decay has no effect on the separation of the individual photon components. Also in the example chosen in Fig.4 is the pulse bandwidth smaller than the EIT bandwidth

Reply: In our simulation results, we can separate the propagation effects resulting from cavity decay and spontaneous emission from the excited state. However, in general we find that the two processes have a qualitatively similar effect on propagation, where both decays effect the separation of the photon components. For example, during propagation a twophoton state can decay via cavity decay (or spontaneous emission) leaving only a single photon in the forward propagating mode. The propagation velocity then changes from the two-photon velocity v_2 to the one-photon velocity v_1 . (We note that the remaining photon still propagates as a dark-state polariton, which may be where the point of confusion arises.) The mix of these two velocities then leads to single photons arriving at the output significantly earlier than if a single photon propagated by itself. In the extreme case where this decay happens just before the two photons would have exited the medium, a single photon is detected with apparent delay time L/v_2 instead of L/v_1 . Depending on the strength of the decay rates, this effect can destroy the separation between detected single photons and bi-photons as we see in our simulations. In the figure below we plot data from trajectories where

I now agree with the authors that also cavity losses will have an effect since after one absorption event from the n -photon component the remaining photons still contribute to the $n-1$ photon wavepacket, however with an effective propagation velocity that is in between the original velocities of the n and $n-1$ components.

Original comment: In contrast to MPS based ground-state methods, the simulation of time-dependent problems using eg. TEBD suffers from the often linear-in-time growth of entanglement entropy. This limits the applicability of TEBD methods to the short time regime. I am missing a more detailed discussion of this issue in the paper. What is the dynamics of the entanglement entropy in the example studied and under what general conditions can one expect that it stays low enough for the method to be applicable to interesting problems

Reply: The scaling of entanglement in a general driven-dissipative system is currently unknown. However, as we discuss in the first paragraph of page 5 of the manuscript there are a number of reasons why we believe MPS will be efficient in treating propagation problems. Specifically, as opposed to a closed system where entanglement can grow indefinitely with time, in photon propagation, photons are continuously leaving the system limiting the entanglement growth. There are then two possible cases: the input is a pulse and the number of photons in the pulse limits entanglement, or continuous driving where the state will eventually reach a steady state with fixed entanglement. In both cases the level of entanglement could still be beyond what is computationally possible, however in practice we are yet to find a situation where that is the case. For pulsed input, as studied in the VIT example, we have added a supplementary document describing how the entanglement entropy scales as the pulse propagates with arbitrary shape in the atomic medium. In this document, we also show that in the case of VIT entanglement entropy remains small.

Here I am not yet fully happy. The estimate given in the supplementary assumes factorizing n -particle wave functions and thus the required bond dimension scales only quadratically in the photon number. While this approximation is good for the considered example, in general the scaling can be much worse. The text in the supplementary suggest that the favorable scaling is the generic case in nonlinear optics. However, e.g. for proposals trying to implement photonic analogues of strongly interacting systems such as the Bose-Hubbard, the Jaynes-Cummings Hubbard model or photonic Laughlin states this is certainly not true. For the photonic strong Tonks gas - considered by one of the authors in Ref. [63] - the required bond dimensions will be much larger than for the example studied in the supplementary. However, this is by no means a fundamental criticism. The key idea of the work is very nice indeed and the manuscript should be published. My suggestion is to revise the text in the supplementary slightly and to add a comment in the main text that MPS methods will reach their limit in the case of strongly interacting photons.

Reviewer #2 (Remarks to the Author):

The authors have made a substantial effort in improving the manuscript and clarify all my previous doubts and concerns. Hence, I fully recommend its publication in Nature Communications.

Reviewer#1's comments:

original comment: "In contrast to MPS based ground-state methods, the simulation of time-dependent problems using eg. TEBD suffers from the often linear-in-time growth of entanglement entropy. This limits the applicability of TEBD methods to the short time regime. I am missing a more detailed discussion of this issue in the paper. What is the dynamics of the entanglement entropy in the example studied and under what general conditions can one expect that it stays low enough for the method to be applicable to interesting problems?"

Reply: The scaling of entanglement in a general driven-dissipative system is currently unknown. However, as we discuss in the first paragraph of page 5 of the manuscript there are a number of reasons why we believe MPS will be efficient in treating propagation problems. Specifically, as opposed to a closed system where entanglement can grow indefinitely with time, in photon propagation, photons are continuously leaving the system limiting the entanglement growth. There are then two possible cases: the input is a pulse and the number of photons in the pulse limits entanglement, or continuous driving where the state will eventually reach a steady state with fixed entanglement. In both cases the level of entanglement could still be beyond what is computationally possible, however in practice we are yet to find a situation where that is the case. For pulsed input, as studied in the VIT example, we have added a supplementary document describing how the entanglement entropy scales as the pulse propagates with arbitrary shape in the atomic medium. In this document, we also show that in the case of VIT entanglement entropy remains small.

Here I am not yet fully happy. The estimate given in the supplementary assumes factorizing n -particle wave functions and thus the required bond dimension scales only quadratically in the photon number. While this approximation is good for the considered example, in general the scaling can be much worse. The text in the supplementary suggest that the favorable scaling is the generic case in nonlinear optics. However, e.g. for proposals trying to implement photonic analogues of strongly interacting systems such as the Bose-Hubbard, the Jaynes-Cummings Hubbard model or photonic Laughlin states this is certainly not true. For the photonic strong Tonks gas - considered by one of the authors in Ref. [63] - the required bond dimensions will be much larger than for the example studied in the supplementary. However, this is by no means a fundamental criticism. The key idea of the work is very nice indeed and the manuscript should be published. My suggestion is to revise the text in the supplementary slightly and to add a comment in the main text that MPS methods will reach their limit in the case of strongly interacting photons.

In response to the reviewers comment we have added additional text to the main text on page 4 clarifying that the bond dimensional scaling may be large for arbitrary photon states: "For arbitrary n -photon wave functions the bound may still scale exponentially in the number of photons $N^{n/2}$, however in the case of vacuum induced transparency that we investigate as a benchmark the scaling is instead approximately quadratic in the number of

photons.” Furthermore, we have added a final sentence to the supplement stating: “For general photon propagation problems, whether the bond dimension required is closer to the bound for the separable case or the non-separable case will be case dependent and determining if MPS treatments remain efficient for photons driven into highly correlated states needs further investigation.”